# Knowledge Transfer in Multi-Task Deep Reinforcement Learning for Continuous Control

**Zhiyuan Xu**[†], **Kun Wu**[†], **Zhengping Che**[‡], **Jian Tang**[†,‡], **Jieping Ye**[‡]

[†]Department of Electrical Engineering & Computer Science, Syracuse University
[‡]DiDi AI Labs, Didi Chuxing
[†]{zxu105, kwu102}@syr.edu
[‡]{chezhengping, tangjian, yejieping}@didiglobal.com

## Abstract

While Deep Reinforcement Learning (DRL) has emerged as a promising approach to many complex tasks, it remains challenging to train a single DRL agent that is capable of undertaking multiple different continuous control tasks. In this paper, we present a Knowledge Transfer based Multi-task Deep Reinforcement Learning framework (KTM-DRL) for continuous control, which enables a single DRL agent to achieve expert-level performance in multiple different tasks by learning from task-specific teachers. In KTM-DRL, the multi-task agent first leverages an offline knowledge transfer algorithm designed particularly for the actor-critic architecture to quickly learn a control policy from the experience of task-specific teachers, and then it employs an online learning algorithm to further improve itself by learning from new online transition samples under the guidance of those teachers. We perform a comprehensive empirical study with two commonly-used benchmarks in the MuJoCo continuous control task suite. The experimental results well justify the effectiveness of KTM-DRL and its knowledge transfer and online learning algorithms, as well as its superiority over the state-of-the-art by a large margin.

## 1 Introduction

The recent breakthrough of Deep Learning (DL) enables Reinforcement Learning (RL) to deliver much better performance in real and complex control tasks. Tremendous successes have been made by Deep Reinforcement Learning (DRL) with Deep Q-Network (DQN) [12] on various discrete control tasks (such as Atari games). Recently, DRL has also been extended to the continuous control domain, which usually leverages an actor-critic based method (such as DDPG [10] and TD3 [4]) to deal with a continuous action space.

Despite the impressive performance of DRL on individual tasks, it remains challenging to train a single DRL agent to undertake multiple different tasks. Unlike single-task DRL, which learns a control policy for an individual task, multi-task DRL requires an agent to learn a single control policy that could perform well on multiple different tasks. Multi-task DRL is considered as an essential step towards Artificial General Intelligence (AGI) [5]. A straightforward approach is to directly train a DRL agent for multiple tasks one by one using a traditional single-task learning algorithm, which has been shown to deliver poor performance and may even fail on some tasks [1, 24], due to differences and possible interference among multiple tasks. Recent research efforts have been made to address the challenges of multi-task DRL. An effective approach is to tackle this problem with knowledge transfer, e.g., Actor-Mimic [14] and policy distillation [15]. These methods usually aimed at training a single multi-task agent under the guidance of task-specific teachers. Through knowledge transfer, one or more control policies can be consolidated into a single one, which can achieve the same level

or sometimes even better performance on individual tasks. However, these methods were designed based on DQN for discrete control tasks. They can not be directly applied to multi-task DRL for continuous control, which usually has a quite different Deep Neural Network (DNN) architecture.

In this paper, we present a Knowledge Transfer based Multi-task Deep Reinforcement Learning framework (KTM-DRL) for continuous control, which enables a single DRL agent to achieve expert-level performance on multiple different tasks by learning from task-specific teachers. In KTM-DRL, the multi-task agent leverages an offline knowledge transfer algorithm designed particularly for the actor-critic architecture to quickly learn a control policy from the experience of task-specific teachers. Then, under the guidance of these knowledgeable teachers, the agent further improves itself by learning from new transition samples collected during online learning. In addition, KTM-DRL leverages hierarchical experience replay for mitigating catastrophic forgetting during both the offline transfer and online learning stages. For performance evaluation, we conduct extensive experiments on two commonly-used MuJoCo benchmarks [19]. The experimental results show that KTM-DRL exceeds the state of the art by a large margin, and its knowledge transfer, online learning, and hierarchical experience replay algorithms turn out to be effective. Particularly, we compare KTM-DRL with an "ideal" solution where the state of the art single-task DRL algorithm for continuous control, TD3 [4], is used to train multiple (instead of one) single-task agents (with much more DNN weights), each of which handles an individual task and is supposed to be the best in its own task. The experimental results show that KTM-DRL can even beat the ideal solution on some tasks and offer very close performance on the others.

## 2 Related Work

**Deep Reinforcement Learning for Continuous Control**   Research efforts have been made to tackle individual continuous control tasks using DRL. A commonly-used approach is the actor-critic based method. Lillicrap *et al.* [10] proposed Deep Deterministic Policy Gradient (DDPG) to learn control policies in high-dimensional and continuous action spaces using the deterministic policy gradient. Haarnoja *et al.* [6] presented Soft Actor-Critic (SAC) based on the maximum entropy RL framework. In SAC, the actor network aims to maximize the expected reward while also maximizing the entropy, which can guide the agent to succeed at the task while acting as randomly as possible for exploration. Furthermore, Fujimoto *et al.* [4] presented an algorithm called Twin Delayed Deep Deterministic policy gradient (TD3), which improves DDPG by employing the minimum value between a pair of critic networks to limit the overestimation introduced by the function approximation errors and using the delayed policy updates to reduce per-update error. Other DRL methods have also been presented for continuous control [13, 16, 17, 20]. All the above works considered the single-task DRL, while this paper targets at multi-task learning.

**Multi-task Deep Reinforcement Learning**   Multi-task DRL has attracted research attention due to its potential for realizing AGI. Parisotto *et al.* [14] presented Actor-Mimic, which leverages the techniques from model compression to train a single multi-task network under the guidance of task-specific teacher networks. Rusu *et al.* [15] proposed the policy distillation, which applies KL divergence to distilling the policy of a DRL agent and trains a new single- or multi-task agent with much smaller DNNs. Liu *et al.* [11] proposed to train a vanilla DQN but with a separate output layer for each task, using all shared hidden layers to learn a common feature representation. Teh *et al.* [18] proposed a shared policy to distill common behaviors from task-specific policies. However, they only evaluated their method on tasks with discrete action space. Yin *et al.* [23] presented a DRL method with a DNN consisting of task-specific convolutional layers and shared multi-task fully-connected layers, which are used to extract unique features from each task and to learn generalized reasoning, respectively. Zhang *et al.* [25] proposed a DRL algorithm that can learn to transfer knowledge from previously-mastered navigation tasks to new problem instances through successor representation learning. These DRL methods were designed particularly for discrete control based on DQN (whose architecture is significantly different from actor-critic networks), which, however, cannot be straightforwardly applied to tackling multi-task continuous control tasks here.

**Multi-task DRL for Continuous Control**   Recent research works presented multi-task DRL methods particularly for continuous control. Arora *et al.* [1] extended the Advantage Actor-Critic (A2C) algorithm by leveraging the policy distillation [15] for handling multi-task continuous control. Carlo *et al.* [2] presented a special DNN to extract a common representation of tasks through a set of shared

layers. Yang *et al.* [21] presented a multi-actor and single-critic DDPG-based architecture to learn multi-tasks. However, without a proper guide from task-specific teachers, a single critic network may fail to learn common knowledge for multiple actors in complex continuous control tasks. Yu *et al.* [24] proposed a form of gradient surgery to avoid interference between tasks. Specifically, it projects the gradient of a task onto the normal plane of the gradient of any other conflicting task, so that the interfering gradient components are not applied. The experimental results presented later show the superiority of our method to others.

# 3 Methodology

## 3.1 Problem Statement

In the typical single-task DRL setting, a DRL agent keeps interacting with an environment in discrete decision epochs. We consider a Markov Decision Process (MDP), in which, at each epoch (i.e., decision step) $t$, the agent observes the current state $\mathbf{s}_t$ and selects an action $\mathbf{a}_t$ based on its control policy $\pi$. After executing the action, the agent receives an immediate reward $r_t$, and the environment transits to a new state $\mathbf{s}'_t$ (a.k.a., $\mathbf{s}_{t+1}$). The agent's goal is to learn a control policy $\pi$ that maximizes the expected return $R = \sum_{i=t}^{T} \gamma^{i-t} r_i$ after $T$ epochs, where $\gamma \in (0, 1]$ is the discount factor. In the multi-task DRL setting, a single agent needs to work simultaneously with multiple environments/tasks to learn the best control policy $\pi$ with the objective of maximizing the total expected return from all environments/tasks. As mentioned above, we target at continuous control where the action space is continuous.

## 3.2 Overview of KTM-DRL

We first give a brief overview of the proposed KTM-DRL, which is illustrated in Figure 1.

We build our DRL agent based on an off-policy actor-critic algorithm called TD3 [4]. The proposed method can be easily modified to be combined with any other off-policy actor-critic algorithm. The actions are real-valued $\mathbf{a}_t \in \mathbb{R}^M$, where $M$ is the dimension of action $\mathbf{a}_t$. TD3 consists of three parameterized DNNs (which are referred to as the DNN triple in the following): an actor network $\pi(\mathbf{s}|\boldsymbol{\theta}^\pi)$ that deterministically maps a state $\mathbf{s}_t$ to an action $\mathbf{a}_t$, and a pair of critic networks $(Q_1(\mathbf{s}, \mathbf{a}|\boldsymbol{\theta}^{Q_1}), Q_2(\mathbf{s}, \mathbf{a}|\boldsymbol{\theta}^{Q_2}))$ that estimate the expected return $R$ (i.e., Q-value) while performing action $\mathbf{a}_t$ in state $\mathbf{s}_t$. Similar to DQN [12], TD3 uses an experience replay buffer $\mathcal{D}$ to store the collected transition samples $(\mathbf{s}_t, \mathbf{a}_t, r_t, \mathbf{s}'_t)$. Suppose that we are given a set of $K$ task-specific expert-level teacher agents $\{\mathcal{T}_1, ..., \mathcal{T}_k, ..., \mathcal{T}_K\}$ (which will be simply called teachers in the following), where teacher $\mathcal{T}_k$ has an actor network $\pi^k(\cdot)$

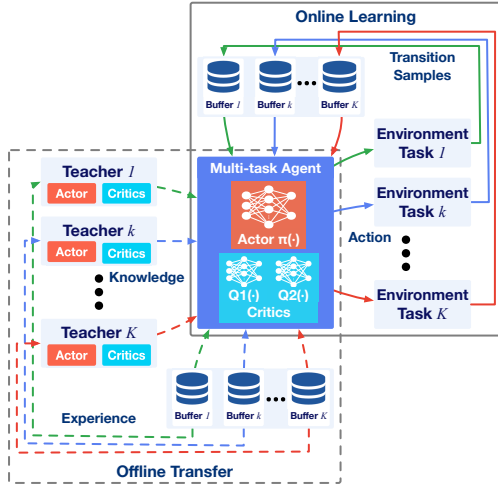

Figure 1: Overview of KTM-DRL

and a pair of critic networks $(Q_1^k(\cdot), Q_2^k(\cdot))$, which are trained with TD3 until convergence on their corresponding tasks $\{\mathcal{M}_1, , ..., \mathcal{M}_k, ..., \mathcal{M}_K\}$, respectively. We aim to transfer the knowledge of both actor and critic networks of those teachers to a single multi-task agent $\mathcal{S}$ such that it can achieve close or even better performance on these tasks.

KTM-DRL consists of two learning stages: the offline knowledge transfer stage and the online learning stage. During the offline knowledge transfer stage, the multi-task agent learns from the experience of task-specific teachers in an offline manner, which is described in detail in Section 3.3. Afterward, during the online learning stage, under the guidance of teachers again, the multi-task agent learns from online transition samples collected by interacting with the environments to further improve its control policy, which is described in detail in Section 3.4.

In addition, to mitigate catastrophic forgetting, in both stages, instead of using a single plain experience replay buffer to store transition samples from all tasks, we create a hierarchical buffer with $K$ separate sub-buffers to store transition samples, each of which corresponds to one task. At each epoch, KTM-DRL samples a mini-batch of $N$ transition samples from each sub-buffer mentioned above and then trains the multi-task agent with every mini-batch. Hence, the agent is trained with $K$ (instead of only one) mini-batches at each epoch. This method helps prevent the multi-task agent from being over-trained on a particular task and forgetting its knowledge on the others. Our ablation study has validated its effectiveness, which is shown in Section 4.1. In KTM-DRL, we use two hierarchical experience replay buffers ($\mathcal{D}_k$ and $\mathcal{D}'_k$) for each task $\mathcal{M}_k$ to store transition samples in the offline transfer and online learning stages, respectively.

### 3.3 Offline Knowledge Transfer

In most of off-policy DRL algorithms, such as DQN [12], DDPG [10], and TD3 [4], transition samples are first stored into an experience replay buffer $\mathcal{D}$, from which mini-batches are sampled for training. Those transition samples are used to train our agent to mimic the teachers' behaviors in an offline and supervised manner without interactions with the environment.

We follow the structure of TD3 with two critic networks to reduce the overestimation bias and improve the stability of the DRL agent. During the offline transfer stage, we jointly optimize the two critic networks of the multi-task agent $\mathcal{S}$ with the following Q-value regression loss function:

$$\mathcal{L}\left(\boldsymbol{\theta}^{Q_1}, \boldsymbol{\theta}^{Q_2}\right) = \frac{1}{N} \sum_{i=1}^{N} \left( \|Q_1^k(\mathbf{s}_i, \mathbf{a}_i) - Q_1(\hat{\mathbf{s}}_i, \hat{\mathbf{a}}_i)\|_2^2 + \|Q_2^k(\mathbf{s}_i, \mathbf{a}_i) - Q_2(\hat{\mathbf{s}}_i, \hat{\mathbf{a}}_i)\|_2^2 \right), \quad (1)$$

where a mini-batch of $N$ state-action pairs $(\mathbf{s}_i, \mathbf{a}_i)$ are sampled from the offline experience reply buffer $\mathcal{D}_k$ of teacher $\mathcal{T}_k$. With Eq. (1), KTM-DRL could quickly mimic the outputs of the two teacher critic networks on each task. Note that the dimensions of the state and action spaces vary from task to task, and thus they cannot be directly fed to the DNN triple of $\mathcal{S}$. Instead of designing a separate DNN triple for each task like those in related works [1, 2, 15, 21], we choose to pad the state and action with appended zeros such that they have fixed lengths. In this way, our agent requires much fewer DNN parameters and training overhead; moreover, it can handle the case with different numbers of tasks. We denote the padded state and action as $\hat{\mathbf{s}}_i$ and $\hat{\mathbf{a}}_i$, respectively. The actor network of the multi-task agent $\mathcal{S}$ is then trained with the gradients computed by:

$$\nabla_{\boldsymbol{\theta}^\pi} \mathcal{G}(\boldsymbol{\theta}^\pi) = \frac{1}{N} \sum_{i=1}^{N} \nabla_{\mathbf{a}} Q_1^k(\mathbf{s}, \mathbf{a})|_{\mathbf{s}=\mathbf{s}_i, \mathbf{a}=f^k(\pi(\hat{\mathbf{s}}_i))} \cdot \nabla_{\boldsymbol{\theta}^\pi} \pi(\mathbf{s})|_{\mathbf{s}=\hat{\mathbf{s}}_i}. \quad (2)$$

To transfer knowledge from teacher $\mathcal{T}_k$ to student $\mathcal{S}$, the first critic network $Q_1^k$ of $\mathcal{T}_k$ is used to calculate the gradient and update the weights $\boldsymbol{\theta}^\pi$. The critic network of teacher $\mathcal{T}_k$ is able to make a precise Q-value estimation for state-action pair $(\mathbf{s}_i, \mathbf{a}_i)$. Therefore, with Eq. (2), it can provide proper guidance for updating the weights of the actor network $\pi$ of agent $\mathcal{S}$. Since the dimension of action varies with tasks, the action predicted by actor network $\pi$ may contain redundant values for $Q_1^k$. Therefore, we apply a slice function $f^k(\cdot)$ to taking only the first $|\mathbf{a}_i|$ values from the predicted action $\pi(\hat{\mathbf{s}}_i)$.

### 3.4 Online Learning

The offline knowledge transfer helps the multi-task agent quickly learn a fairly good control policy from the teachers. However, without interacting with the actual environments, the multi-task agent cannot gain sufficient new knowledge and thus may lead to over-fitting. Hence, we propose an online learning algorithm that enables the agent to further improve its control policy with newly collected online transition samples. During the online learning stage, instead of learning from task-specific teachers, the agent updates its critic networks with TD-errors, which is similar to the training procedure of TD3 [4]. The target value $y_i$ for training the critic networks is given as:

$$\begin{aligned} y_i &= r_i + \gamma \min_{j=1,2} Q'_j \left( \hat{\mathbf{s}}'_i, g^k \left( \pi'(\hat{\mathbf{s}}'_i) \right) + \epsilon \right), \\ \epsilon &\sim \text{clip}\left( \mathcal{N}(0, \sigma^2), -c, c \right), \end{aligned} \quad (3)$$

where $\pi'(\cdot)$ and $Q'_j(\cdot)$ $(j = 1, 2)$ are the target networks of the actor and critic networks, respectively, which is used for stabilizing the training process [10, 12], $\gamma$ is the reward discount factor, and $\epsilon$ is the

clipped exploration noise, which is sampled from the normal distribution $\mathcal{N}(0, \sigma^2)$ and clipped by a threshold $c$. Unlike DQN [12] and DDPG [10], to avoid the bias introduced by the policy update, TD3 [4] leverages two (instead of one) critic networks $Q_1$ and $Q_2$ for making a better estimation for the Q-value. Note that to concentrate on the action values particularly for task $k$, we apply a mask function $g^k(\cdot)$ to filtering out redundant values by setting them to 0 for the output of the actor network, which is especially helpful when the tasks have different action dimensions. Thereafter, the two critic networks are trained by the following loss function:

$$\mathcal{L}\left(\boldsymbol{\theta}^{Q_1}, \boldsymbol{\theta}^{Q_2}\right) = \frac{1}{N}\sum_{i=1}^{N} \|y_i - Q_1(\hat{\mathbf{s}}_i, \hat{\mathbf{a}}_i)\|_2^2 + \|y_i - Q_2(\hat{\mathbf{s}}_i, \hat{\mathbf{a}}_i)\|_2^2. \tag{4}$$

Unlike TD3, the actor network of the agent is then trained under the guidance of both its critic network and teachers:

$$\begin{aligned}
\nabla_{\boldsymbol{\theta}^\pi} \mathcal{G}(\boldsymbol{\theta}^\pi) = {} & \frac{\alpha}{N}\sum_{i=1}^{N} \nabla_{\mathbf{a}} Q_1^k(\mathbf{s}, \mathbf{a})|_{\mathbf{s}=\mathbf{s}_i, \mathbf{a}=f^k(\pi(\hat{\mathbf{s}}_i))} \cdot \nabla_{\boldsymbol{\theta}^\pi} \pi(\mathbf{s})|_{\mathbf{s}=\hat{\mathbf{s}}_i} \\
& + \frac{\beta}{N}\sum_{i=1}^{N} \nabla_{\mathbf{a}} Q_1(\mathbf{s}, \mathbf{a})|_{\mathbf{s}=\hat{\mathbf{s}}_i, \mathbf{a}=g^k(\pi(\hat{\mathbf{s}}_i))} \cdot \nabla_{\boldsymbol{\theta}^\pi} \pi(\mathbf{s})|_{\mathbf{s}=\hat{\mathbf{s}}_i},
\end{aligned} \tag{5}$$

where the first term is the same as that used in the offline transfer stage, which we leverage for continuing the knowledge transfer from the teachers. We add the second term to compute the

---

**Algorithm 1:** KTM-DRL

**Input:** The number of offline transfer and online learning epochs $T_{\text{off}}$ and $T_{\text{on}}$, the policy update frequency $d$, a set of $K$ tasks, a set of $K$ teachers and each teacher $\mathcal{T}_k$ with its DNN triple $(\pi^k, Q_1^k, Q_2^k)$, a multi-task agent $\mathcal{S}$ with its DNN triple $(\pi, Q_1, Q_2)$ and their weights $(\boldsymbol{\theta}^\pi, \boldsymbol{\theta}^{Q_1}, \boldsymbol{\theta}^{Q_2})$, and the corresponding target networks $(\pi', Q_1', Q_2')$ with weights $(\boldsymbol{\theta}^{\pi'}, \boldsymbol{\theta}^{Q_1'}, \boldsymbol{\theta}^{Q_2'})$;

1   Randomly initialize the weights $(\boldsymbol{\theta}^\pi, \boldsymbol{\theta}^{Q_1}, \boldsymbol{\theta}^{Q_2})$ of the DNN triple of $\mathcal{S}$;
    /**Offline Knowledge Transfer**/
2   **while** *decision epoch $t < T_{off}$* **do**
3      **foreach** *task $k$* **do**
4         Sample $N$ transition samples $(\mathbf{s}_i, \mathbf{a}_i, r_i, \mathbf{s}_i')$ from offline experience replay buffer $\mathcal{D}_k$;
5         Update critic networks $Q_1$ and $Q_2$ with the loss function given by Eq. (1);
6         Update actor network $\pi$ with the gradient calculated by Eq. (2);
7      **end**
8   **end**
9   Synchronize the weights of the target networks $\boldsymbol{\theta}^{\pi'} := \boldsymbol{\theta}^\pi$, $\boldsymbol{\theta}^{Q_1'} := \boldsymbol{\theta}^{Q_1}$, $\boldsymbol{\theta}^{Q_2'} := \boldsymbol{\theta}^{Q_2}$;
    /**Online Learning**/
10   **while** *decision epoch $t < T_{on}$* **do**
11      **foreach** *task $k$* **do**
12         Select an action with exploration noise $\epsilon \sim \mathcal{N}(0, \sigma^2)$: $\mathbf{a}_t := \pi(\mathbf{s}_t) + \epsilon$;
13         Execute action $\mathbf{a}_t$, receive reward $r_t$, and observe new state $\mathbf{s}_t'$;
14         Store transition sample $(\mathbf{s}_t, \mathbf{a}_t, r_t, \mathbf{s}_t')$ into online experience replay buffer $\mathcal{D}_k'$;
15         Sample $N$ transition samples $(\mathbf{s}_i, \mathbf{a}_i, r_i, \mathbf{s}_i')$ from online experience replay buffer $\mathcal{D}_k'$;
16         Calculate the target value by Eq. (3);
17         Update critic networks $Q_1$ and $Q_2$ with the loss function given by Eq. (4);
18         **if** *$t$ mod $d$ = 0* **then**
19            Update actor network $\pi$ with the gradient calculated by Eq. (5);
20            Update the weights of target networks $\boldsymbol{\theta}^{\pi'} := \tau\boldsymbol{\theta}^\pi + (1-\tau)\boldsymbol{\theta}^{\pi'}$, $\boldsymbol{\theta}^{Q_1'} := \tau\boldsymbol{\theta}^{Q_1} + (1-\tau)\boldsymbol{\theta}^{Q_1'}$, $\boldsymbol{\theta}^{Q_2'} := \tau\boldsymbol{\theta}^{Q_2} + (1-\tau)\boldsymbol{\theta}^{Q_2'}$;
21         **end**
22      **end**
23   **end**

---

gradients from its critic network for improving its control policy, just like that in a regular actor-critic algorithm. The two parameters $\alpha$ and $\beta$ are used to express the relative importance of the two parts.

We summarize KTM-DRL in Algorithm 1. In our implementation, the key hyper-parameters were set as follows: $\alpha = \beta = 1$, the exploration noise is $\mathcal{N}(0, 0.1)$, the clip threshold $c = 0.5$, and the policy update frequency $d = 2$. The size of each replay buffer $|\mathcal{D}_k|$ and $|\mathcal{D}'_k|$ and a mini-batch is set to $10^6$ and 256, respectively. Reward discount factor $\gamma$ is set to 0.99, target network update rate $\tau$ is set to 0.005, and the learning rate for both actor and critic networks is set to $3 \times 10^{-4}$. In KTM-DRL, every DNN consists of only 2 hidden layers with 400 ReLU activated neurons each. While we believe a fine-grained tuning of settings for each teacher (e.g., adjusting the number of layers, the number of neurons, and hyper-parameters) may lead to better performance on its task, we just followed up the standard settings of TD3 [4] for all teachers in our experiments.

# 4 Performance Evaluation

To evaluate the proposed KTM-DRL, we conducted extensive experiments with the continuous control tasks in the MuJoCo suite [19]. We employed two typical benchmarks (which will be called Benchmarks A and B in the following): 1) Benchmark A: it is called the *HalfCheetah* task group [7], which includes 8 similar tasks; 2) Benchmark B: it consists of 6 considerably different tasks.

For a comprehensive evaluation, we compared KTM-DRL with an "ideal" solution, the extension of the state of the art single-task DRL methods, as well as the state of the art multi-task DRL algorithms proposed in recent papers. Specifically, as mentioned above, the ideal solution leverages TD3 to train $K$ (instead of one) single-task agents (each corresponds to an individual task), which can serve as an upper bound. We extended TD3 [4] (TD3-MT) and SAC [6] (SAC-MT) to train a single-task DRL agent for multiple tasks one by one. Note that for fair comparisons, we applied the above hierarchical

| Method | KTM-DRL (ours) | Ideal | TD3-MT [4] | SAC-MT [6] | SharedNet [2] | G-Surgery [24] | A2C-MT [1] |
|---|---|---|---|---|---|---|---|
| HCSmallTorso | **10348** | 8743 | 7898 | 7805 | 4140 | 7189 | 2276 |
| HCBigTorso | **10364** | 9067 | 9075 | 7944 | 1463 | 7737 | 1428 |
| HCSmallLeg | **10594** | 9575 | 9065 | 8026 | 4885 | 7564 | 1525 |
| HCBigLeg | 10402 | **10683** | 7982 | 7933 | 1990 | 7275 | N/A |
| HCSmallFoot | 8836 | **9633** | 7718 | 7284 | 3752 | 7197 | 1480 |
| HCBigFoot | **9239** | 8902 | 6496 | 7340 | 4661 | 6826 | 2188 |
| HCSmallThigh | 10470 | **10769** | 8053 | 7844 | 4906 | 7240 | 1874 |
| HCBigThigh | **9787** | 9524 | 8867 | 7579 | 3547 | 7501 | N/A |

Table 1: The max average rewards on Benchmark A

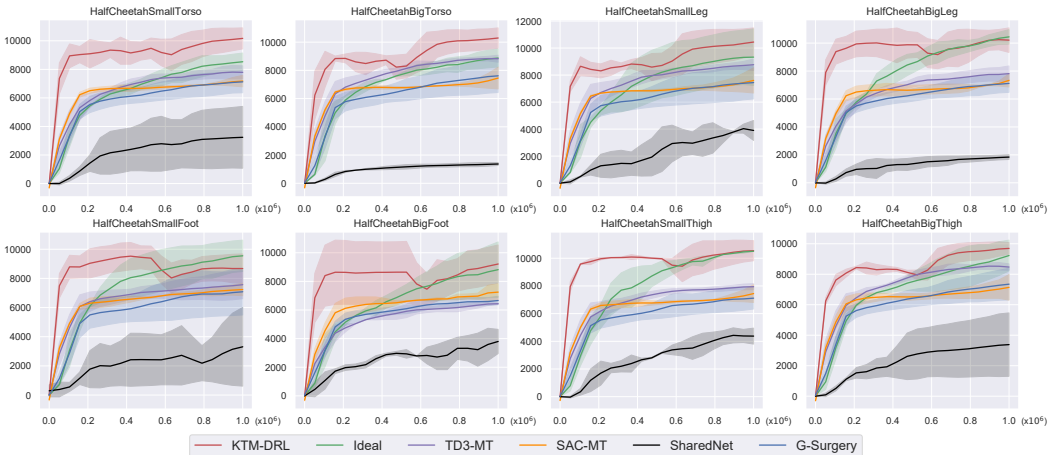

Figure 2: The learning curves on Benchmark A

| Method | KTM-DRL (ours) | Ideal | TD3-MT [4] | SAC-MT [6] | SharedNet [2] | G-Surgery [24] |
|---|---|---|---|---|---|---|
| Ant | 5836 | **5839** | 324 | -31 | 885 | 399 |
| Hopper | 3565 | **3588** | 2464 | 2249 | 1006 | 544 |
| Walker2d | **4863** | 4797 | 3957 | 2672 | 2970 | 733 |
| HalfCheetah | 10921 | **10969** | 4826 | 908 | 4404 | 2310 |
| InvPendulum | **1000** | 1000 | 1000 | 1000 | 1000 | 12 |
| InvDbPendulum | 9347 | 9351 | **9358** | 5358 | 9341 | 4332 |

Table 2: The max average rewards on Benchmark B

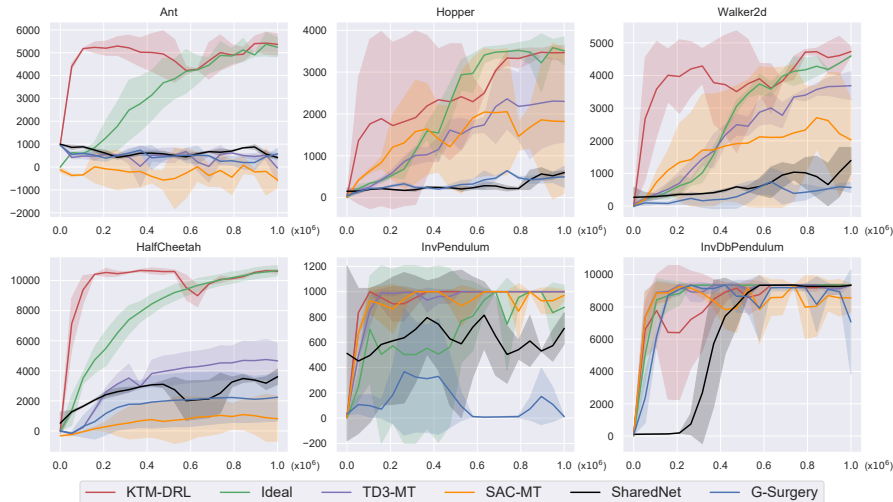

Figure 3: The learning curves on Benchmark B

experience replay and the mask function in Eq. (3) to TD3-MT and SAC-MT. Otherwise, these two methods turned out to fail on some of the tasks. The other baselines include a recent shared network based algorithm (SharedNet) [2], another recent method based on gradient surgery (G-Surgery) [24], and a multi-task DRL algorithm based on Advantage Actor-Critic (A2C) [1](A2C-MT). Note that G-Surgery is claimed to be compatible with any actor-critic based DRL algorithm. In the original paper [24], the authors implemented it based on SAC [6]. Considering TD3 generally outperforms SAC on the MuJoCo continuous control tasks, for fair comparisons, we implemented G-Surgery based on TD3. Due to no public code available, we implemented Distral [18] and multi-task DDPG [21], and they always achieve negative rewards on most of the tasks on Benchmark B. Thus we omitted their results.

We trained each method for 1 million decision epochs and then evaluated it for 10 trials to report the max average rewards per episode. In KTM-DRL, we trained the multi-task agent for 500K epochs in both the offline transfer and online learning stages. We had 2 independent runs for all methods. Note that due to the lightweight design of our framework, the training complexity of KTM-DRL is almost the same as directly extending TD3 to multiple tasks (i.e., TD3-MT), which is much less than other methods. More specifically, it takes about 12 hours for KTM-DRL to finish the 1M end-to-end training with NVIDIA Tesla P100 GPU on Benchmark A, comparing to about 5.5 hours, 11.5 hours, 25 hours, and 31 hours for ideal (single task), TD3-MT, SAC-MT, and G-Surgery, respectively. The results on the two benchmarks are shown in Table 1 and Table 2, respectively. Moreover, the corresponding learning curves are shown in Figure 2 and Figure 3, in which the x-axis represents the total number of training epochs (steps) and the y-axis shows the achieved rewards. The shaded region the standard deviation of the average rewards over different runs. Based on these results, we can make the following observations:

1) Even though KTM-DRL has much fewer DNN weights than the ideal solution, it can even beat the ideal solution on some tasks while offering very close performance on the others. For example,

on Benchmark A, KTM-DRL leads to $18\%$ and $14\%$ improvements over the ideal solution on tasks *HalfCheetahSmallTorso* and *HalfCheetahBigTorso*, respectively. On average (over all the tasks), KTM-DRL beats the ideal solution by $2\%$, which validates its effectiveness. It is worth to mention that related works [3, 14, 15] also made similar observations that a single multi-task agent may surpass their task-specific teachers. One motivating argument [3] is that the distillation encourages the agent to explore more states than teachers and thus could potentially lead to a better performance.

2) KTM-DRL consistently outperforms all the other baselines on almost all tasks of both two benchmarks. On average, KTM-DRL achieves $42\%$, $110\%$, $130\%$, and $180\%$ improvements over TD3-MT, SAC-MT, SharedNet, and G-Surgery, respectively. It is interesting to see that SharedNet and G-Surgery perform poorly on several tasks of Benchmark B. This demonstrates their weakness on considerably diverse tasks. In this case, it may be hard to directly learn a high-quality representation from scratch for knowledge sharing, which, however, determines whether SharedNet or G-Surgery can obtain high scores. Moreover, as expected, these results show that a straightforward extension of a single-task DRL agent (such as TD3-MT and SAC-MT) does not offer a satisfying performance (even with the help of the hierarchical experience replay).

3) From the learning curves shown in Figure 2 and Figure 3, we can observe that the multi-task agent of KTM-DRL quickly (after around 50K epochs) learns how to do well on different tasks during the offline knowledge transfer stage. Then, the multi-agent is further improved during the online learning stage. This observation well illustrates the necessity and effectiveness of both the offline knowledge transfer and the online learning.

## 4.1 Ablation Study

We performed a comprehensive ablation study for the three key components of KTM-DRL, including the offline knowledge transfer (Section 3.3), the online learning (Section 3.4), and the hierarchical experience replay, on Benchmark B, to further understand their effectiveness and contributions. Similar to above, for fair comparisons, we trained each method for 1 million epochs and then evaluated it for 10 trials. For a better representation, we used the ratio of the max averaged rewards given by a method to that of the ideal solution as the performance metric, which we call *performance ratio* in the following.

| Method | KTM-DRL (ours) | w/o Offline | w/o Online | Online w/o Teacher | Online Only Teacher | w/o HER |
|---|---|---|---|---|---|---|
| Ant | 0.9745 | 0.9767 | 0.9035 | 0.9630 | **0.9784** | 0.2215 |
| Hopper | **1.0437** | 0.8977 | 0.9761 | 0.9715 | 0.9836 | 0.9406 |
| Walker2d | **1.0112** | 0.9493 | 0.9035 | 0.9872 | 0.7694 | 0.8961 |
| HalfCheetah | **1.0080** | 0.9187 | 0.9784 | 0.9752 | 0.9832 | 0.8274 |
| InvPendulum | **1.0** | 1.0 | 0.9542 | 1.0 | 1.0 | 1.0 |
| InvDbPendulum | 0.9996 | 0.9991 | 0.9974 | **1.0016** | 0.9992 | 0.9997 |

Table 3: The performance ratio given by different methods in the ablation study

1) To validate the contribution of the proposed offline knowledge transfer algorithm, we trained a multi-task agent without the offline transfer. The solution is labeled as "w/o Offline". We can see from these results that the offline knowledge transfer leads to $5\%$ gain in terms of performance ratio on average.

2) To verify the effectiveness of our online learning algorithm, we considered three different online learning schemes: a) Training the multi-task agent without the online learning, which is labeled as "w/o Online". b) Training the agent with the online learning but without the help of teachers, which is labeled as "Online w/o Teachers". In this case, we modified Eq. (5) to remove the term related to teachers. c) Training the agent with the online learning but with only the guidance of teachers, which is label as "Online only Teacher". In this case, we modified Eq. (5) to only keep the term related to teachers. As shown in Table 3, KTM-DRL is superior to these three schemes on almost all the tasks; and they result in $5\%$, $2\%$, and $5\%$ losses respectively in terms of performance ratio on average.

3) In addition, we investigated the effectiveness of the hierarchical experience replay (HER). We compared KTM-DRL with another commonly-used training method [15], which keeps training

an agent with transition samples from one task for a whole episode and then switches to another. We labeled this method as "w/o HER" in Table 3. We can observe from the results that KTM-DRL performs consistently better than KTM-DRL w/o HER on all the tasks, and on average, the hierarchical experience replay leads to $23\%$ improvement in terms of the performance ratio.

## 5   Conclusions

In this paper, we presented KTM-DRL, which enables a single multi-task agent to leverage the offline knowledge transfer, the online learning, and the hierarchical experience replay for achieving expert-level performance in multiple different continuous control tasks. For performance evaluation, we conducted a comprehensive empirical study on two commonly-used MuJoCo benchmarks. The extensive experimental results show that 1) KTM-DRL beats the ideal solution with much more DNN weights on some tasks while offering very close performance on the others; 2) KTM-DRL outperforms the state of the art by a large margin; and 3) the offline knowledge transfer algorithm, the online learning algorithm, and the hierarchical experience replay are indeed effective. Note that we assume all the given teachers have expert-level performance, or KTM-DRL may fail to find a good multi-task control policy. As far as we know, it remains an open question whether we can learn policies from imperfect or sub-optimal teachers. This challenging problem is beyond the scope of our discussion, and we leave it for future work.

## Acknowledgements and Funding Disclosure

We would like to thank our attentive anonymous AC and reviewers for their useful comments that have greatly improved this manuscript. This work was done while the authors, Zhiyuan Xu and Kun Wu, were interns at DiDi AI Labs.

## Broader Impact

This work presents a multi-task learning framework for training a single (instead of multiple) DRL agent to undertake multiple different tasks. Even though there is still a long way to go, we believe this work may lead us one step closer to Artificial General Intelligence (AGI). The proposed algorithm may be used in a large variety of domains (such as transportation, medical care, home services, and manufacturing) to train a machine or robot to master many skills and handle different tasks. This work and the corresponding applications may reduce workload and improve the quality of life for human beings with much less computing and energy resources (which could potentially benefit the environment too). In addition, our task/method does not leverage biases in the data. However, at the same time, this kind of technology may have some negative consequences because when a machine or robot is able to gain more skills or capabilities than a human, then we may have to face losses of jobs. What is more, by using our framework, a machine could potentially learn unexpected skills from malicious teachers, and this increases the risk of technology being used incorrectly and hazardously. Therefore, we should pay careful attention to multi-task learning for machines and robots.

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
