[Reviews · NeurIPS 2020]

Review 1

Summary and Contributions: The authors introduce KTM-DRL, an algorithm to leverage knowledge acquired by a set of teachers in different tasks into a single-actor network, targeting for continuous control.

Strengths: I particularly liked that the paper includes various experiments and an ablation study to reflect the contributions of each of the novel components of the proposed approach.

Weaknesses: The significance of the contribution is kind of limited since it requires to train a set of actors for each learning task before training the actual algorithm. The experimental section needs to better explain why the results happen, some of them were surprising and authors do not provide needed explanations (I detail more about this below). The theoretical explanation needs further clarification (more below).

Correctness: The empirical results, include an "ideal" training network, where the proposed algorithm is compared as an single-task algorithm trained on that task which authors themselves claim to be intended to work as an upper bound. I think this work should detail why the proposed approach seems too be working better even than the upper bound. How the limits on performance of the teachers impact on the performance of the proposed approach? i.e. if the teacher for a given task has converged to a suboptimal policy, does KTM-DRL converge to that policy too? The ideal solution has much more DNN weights, couldn't it work better with less?

Clarity: Background section of the basic concepts needed by the reader? what kind of model are you employing for the environment, an MDP? Line 150, what do you exactly mean by saying that your agent is trained in a supervised manner? Equations such as (1) or (2) should have intuitive explanations. Redaction in Section 3.4 and 5, need rework and proof reading.

Relation to Prior Work: The authors do an extensive contrast with previous contributions. However, since the authors claim that their approach outperforms all the previous works, why was the work [19], using a multi-task DDPG not included as a baseline?

Reproducibility: No

Additional Feedback: About reproducibility, I particularly missed the learning rate, hierarchical replay buffer sizes and the specific number of independent runs for the training plots. The broader impact section feels to speculative and broad, I would rather try to be more specific about the impacts given by the contributions of this work, rather than the impact of the field in general. The last statement, needs further clarification and explanation. Some minor comments: Line 158 "cases" instead of "the cases" Line 214, what are exactly "decision epochs"? --- Post-Rebuttal comments: I appreciate the effort authors have done to address my concerns given the space available, I feel overall satisfied with your answers and have consequently increased my score. I suggest to include in the final version as many of those clarifications as possible, specially why benchmark [19] is not present and why sometimes the results are better than the "ideal solution".


Review 2

Summary and Contributions: The paper proposes a multi-task RL method for continuous control tasks using single task experts in 2 separate training steps. The method builds on TD3. In the first training stage which is offline from the perspective of the main multitask agent, it is being trained to imitate data from the single task experts without interacting with the environment (actor and critic networks). In the second step, the network itself is acting in the environment to fill a replay buffer, but training builds on multiple replays including the new replay and replays for each expert. The approach performs well on a couple of variations from OpenAI’s gym tasks. After reading the author feedback I sadly do not see myself able to change the review. Additional information such as of the pretraining for experts is appreciated but the overall criticism remains. The approach mostly gains from having pretrained experts. Therefore additional data is required and the use of experts per task is commonly known (see 'Distral: Robust Multitask Reinforcement Learning' or 'Distilling Policy Distillation' for an overview). If the particular way of using experts is the main contribution, then it has to be compared against other approaches which use experts. While an interesting perspective and the authors should continue to improve the submission, this paper - as is - does not meet the standards for NeurIPS from my point of view.

Strengths: The structure of the paper is clear. In the test domains and given a set of intuitive baselines, the method performs commensurate or better. The explicit 2 step training procedure is interesting and opens up new questions for future work. The ablation study 4.1 provides insights into which aspects are relevant for the method.

Weaknesses: Novelty of the proposed method is limited as training from experts to accelerate training is common across many prior approaches (also in continuous action spaces) [1,2,3] (these three only serve as examples and there are likely more advanced methods published in the last years), and many aspects of the training framework are given in ActorMimic for the discrete action case (also referred to in the paper). If this particular way of using experts during training creates significant benefits over existing approaches, this needs to be benchmarked (e.g. against [1,2,3]) to strengthen the contributions. (the ablation against aspects of the introduced method alone does not suffice here) The current training of the experts is ambiguous from the paper. Are the experts pretrained or are they trained in parallel?( however, even parallel training would give more training data than was available for the other methods if it is not currently counted) There are some repeated grammatical error in the paper (e.g. additional or missing pronouns). [1] Vecerik, Mel, et al. "Leveraging demonstrations for deep reinforcement learning on robotics problems with sparse rewards." arXiv preprint arXiv:1707.08817 (2017). [2] Zhu, Yuke, et al. "Reinforcement and imitation learning for diverse visuomotor skills." arXiv preprint arXiv:1802.09564 (2018). [3] Sun, Wen, J. Andrew Bagnell, and Byron Boots. "Truncated horizon policy search: Combining reinforcement learning & imitation learning." arXiv preprint arXiv:1805.11240 (2018).

Correctness: Method and evaluation seem generally correct.

Clarity: Overall structure of the paper is clear and intuitive. See weaknesses: There are some repeated grammatical error in the paper (e.g. additional or missing pronouns).

Relation to Prior Work: Connection to some fundamentals of RL (DQN, DDPG, TD3) are done well. Connections to other work in multitask learning and learning from demonstration is very limited and requires to be improved (similarly the earlier mentioned work combining RL/IL)

Reproducibility: Yes

Additional Feedback: Describing RL as ‘de facto’ approach to complex tasks could be phrased a bit more humble. Many other approaches address ‘complex tasks’ and even if we limit ourselves to continuous control tasks, there is a considerable community working on optimal control which should not be ignored. Similarly there is significant work on multitask RL such that ‘scant attention has been paid’ is in part incorrect. Using the name ‘‘ideal’ solution” for independent solution of individual tasks is incorrect as no transfer can happen between the tasks which can improve performance. Given the description mentions that TD3 uses 2 critic networks, it would be helpful to mention their purpose for more consistency. Since the main benefits happen mostly early during training, final performance is unsuited for the ablation. It would be better to either use training curves or any metrics which integrate over the training procedure. (General hint:) The method to ‘pad the state and action with appended zeros such that they have fixed lengths’ might potentially not the best idea as these states might have different effects and semantics across tasks. I would recommend to potentially introduce one task specific layer to map towards an embedding which has the same size across tasks. (General hint2:) I can highly recommend [4] for challenges in imitation learning and the issue of compounding errors. [4] Ross, Stéphane, Geoffrey Gordon, and Drew Bagnell. "A reduction of imitation learning and structured prediction to no-regret online learning." Proceedings of the fourteenth international conference on artificial intelligence and statistics. 2011.


Review 3

Summary and Contributions: This paper proposes a policy distillation mechanism in the context of multi-task learning. First, expert policies are trained in a single-task setting for each task in a collection of tasks. Next, these individual policies are distilled into a meta-policy using a Q-value regression loss. Finally, the meta-learner is fine-tuned via direct exploration in the multi-task setting. The paper also introduces additional optimization knobs like a segregated experience replay buffer and a unified actor/critic architecture for all tasks via state- and action-space padding.

Strengths: The paper is clearly written, motivates the problem well and was a delight to read. Comparisons with prior work are thorough and care was taken to ensure that auxiliary optimizations like the segregated replay buffer was implemented in the baselines for a fair comparison. The ablation studies were also rigorous and demonstrated the efficacy of individual components of KTM-DRL. Finally, the results are strong and demonstrate clear advantages over current state-of-the-art multi-task methods like PCGrad (aka gradient surgery for multitask learning).

Weaknesses: One aspect of the paper gives me pause. It is not clear what the x-axis of the learning curves represent. This should ideally be the total number of steps - where each step involves a single interaction with the environment. It appears that the authors are plotting "epochs" which does not capture the sample-efficiency of the approaches. Secondly, it is not clear if the x-axis included all of the training steps involved in the offline training of the teacher agents cumulatively. This is essential in order to make a fair comparison to other baselines that do not require offline training. Finally, a discussion on the relative complexity of KTM-DRL and other online methods is also missing - it is not clear what the end-to-end training time would be on comparable hardware for an online method vs KTM-DRL. In order to have a fair comparison, KRM-DRL should report

Correctness: Yes. Yes.

Clarity: Yes.

Relation to Prior Work: Yes.

Reproducibility: Yes

Additional Feedback: I am currently scoring this paper based on the assumption that the x-axis in the learning curves are *not* cumulative samples across all offline/online learning for KTM-DRL. I am assuming that for KTM-DRL, these are only the online steps. if the authors confirm otherwise, I would be willing to improve my score. Post rebuttal comments: Thanks to the authors for addressing mine and other reviewers' questions in the rebuttal. Based on that I am increasing my score by 1 point. I agree with other reviewers that the verbiage needs to be adjusted in the paper. Also, please include the suggested prior work - e.g., Distral - that are in the intersection of policy distillation and multi-task or meta-learning.


Review 4

Summary and Contributions: This paper proposes a novel multi-task RL framework that can train a single agent to handle multiple different continuous control tasks. In this framework, the multi-task agent first distills a policy from task-specific teachers by an offline knowledge transfer algorithm, and then it uses an online learning algorithm to fine-tune the learned policy through interacting with environments for all tasks. The agent will also keep mimicking the offline teachers during the online tuning. They conduct experiments on two MuJoCo benchmarks to verify the effectiveness of their proposed methods. Contributions: 1. This paper proposes a new framework that combines offline policy distillation with online policy turning to improve multi-task RL for continuous control. 2. The proposed method significantly outperforms baseline methods and achieve similar performance with “ideal” agent (i.e., train individual agent for each task). The authors also conduct ablation study to verify the contributions of every components of their framework.

Strengths: 1. This paper tackles a very valuable problem of learning a single agent for all tasks, which is an important step towards Artificial General Intelligence. The idea of combining offline policy distillation with online policy turning is interesting and novel. This paper is relevant to a broad range of researches on multi-task RL. 2. The proposed multi-task RL framework can achieve comparable performance with training individual agent for each task. The authors provide ablation study to verify the effectiveness of each component.

Weaknesses: Some experimental details are missed. How many seeds are used? What’s the variance of the reported scores in Table 1, 2 and 3?

Correctness: The claims and method are correct. The empirical methodology is sound.

Clarity: The paper is well-written.

Relation to Prior Work: Yes.

Reproducibility: Yes

Additional Feedback: 1. I am not clear about the concrete design of hierarchical experience replay. Maybe more descriptions on hierarchical experience replay are necessary. 2. Can you give some failed examples and summarize when and why your algorithm will not work well 3. Are there any implementation codes? ========================================== post-rebuttal comment: I have read all reviews and the response of authors. I retain my score since most of my concerns are addressed. In addition, I agree with Reviewer #2 that it is necessary to compare with imitation learning and multi-task learning approaches that uses experts trained by RL on the same dataset.

[Author Response · NeurIPS 2020]

# Responses to Review Comments (Paper #1720)

We thank all the reviewers for their valuable and helpful comments. Our responses are presented as follows:

**Responses to Reviewer #1** 1. Weaknesses and Correctness: 1) We believe the novelty of our work lies on that we
propose a framework that enables a single DRL agent to achieve expert-level performance on multiple different tasks by
learning from task-specific teachers. In general, it is much easier to obtain a set of actors for each learning task; 2) In
some scenarios KTM-DRL works better than the ideal solution. Related works [13] (Actor-Mimic) and [14] (policy
distillation) also made such observation. One motivating argument from [24] is that the distillation encourages the agent
to explore more potential states than teachers, thus leading to better performance ([24] Czarnecki et al., *Distilling Policy*
*Distillation*). 3) In KTM-DRL, we assume a teacher that has good performance on the task will be given. As far as we
know, it remains an open question that whether we can learn policies from in-perfect export or sub-optimal policies
(Imitation Learning). This problem is quite challenging and is beyond the scope of our discussion, we leave it for future
work; 4) While we believe fine-grained tuning to neural network settings (e.g., reducing the number of layers and the
number of neurons) may lead to better performance on its task, we just followed up standard settings with previous
work [3] (TD3).

2. Clarity, Relation, additional Comments: 1) We follow a typical setting for the DRL problem, which is a MDP; 2) For
the term "supervised manner" used in Line 150, we want to emphasize that the DRL agent updates its policy only based
on a pre-collected dataset without interactions with the environment; 3) For Equations 1 and 2, "We follow the structure
of two critic networks as TD3, to reduce the overestimation bias and improve the stability of the algorithm. In Equ. 1,
we aim at mimic the outputs of two critic networks of each task-specific teacher. In Equ. 2, we use one of the critic
network of teacher to estimate a precise Q-value, which can provide proper guidance for updating the actor network of
the student."; 4) Since we can not find any public implementation of the multi-task DDPG [19], we implemented it
and evaluate it by ourselves. However, we found its performance is quite bad. For example, on the benchmark B, the
multi-task DDPG [19] only achieves negative rewards on all the tasks. Thus we omit the results of multi-task DDPG; 5)
We set the learning rate to $3 \times 10^{-4}$ for both actor and critic network, the total buffer size is set to $1 \times 10^6$. We have
two independent runs for the training plots. We will narrow down the scope and discuss more specifically in the broader
impact section. Besides, we will have a further proofreading and fix the typos.

**Responses to Reviewer #2** 1. Related works: 1) We thank you for the suggestion to compare with Imitation Learning
(IL) methods [1, 2, 3]. But we respectfully disagree with you. The major goal of our paper is multi-task learning, i.e.,
train a single DRL agent to achieve expert-level performance in multiple different tasks. However, these IL methods
focus more on how to leverage demonstrations to effectively learn the control policy for the same task. Specifically,
none of [1, 2, 3] on RL+IL are for multi-task with single agent. It is also not trivial to extend existing methods into the
continuous-control multi-task DRL setting, which is demonstrated by our experiments (i.e., TD3-MT and SAC-MT can
not work well on the multi-task setting). 2) Thanks for your suggestion about the references, we will certainly take a
step further to investigate more related works. But it is good to mention that we already include some most related
works (i.e., multi-task DRL for continuous control tasks) and compare with them in our experiments.

2. Others: 1) The expert are pre-trained. In general, it is much easier to train an expert on a single task, there are quite a
few DRL methods for continuous control that can achieve state-of-the-art performance, like DDPG, TD3, SAC, and
PPO. Thus in this paper, we didn't discuss much about how to train the expert on a single task, which is beyond the
scope of our discussion. 2) Thanks for your suggestion on wording. We will replace them with more accurate words in
the paper and have a further proofreading. 3) We will add more descriptions about the two critic networks in the paper.

**Responses to Reviewer #3** We would like to thank the reviewer for the positive comments and encouragement. For
the questions: 1) Sorry for the confusion. In the paper, we use the term "epoch" to represents one single interaction
with the environment, which has the same meaning of one "step". Thus the x-axis of the learning curves represent the
total number of steps. 2) The x-axis in the learning curves **are** cumulative samples across all offline/online learning for
KTM-DRL. We will emphasize this in the paper. 3) Thanks for your suggestion. We will add more discussion in the
paper. Basically, KTM-DRL actually takes much less time to finish the 1M training compared with some baselines, e.g.,
it takes 19 hours to finish the end-to-end training, compared with 26 hours for SAC-MT and 28 hours for G-Surgery on
NVIDIA GTX 960.

**Responses to Reviewer #4** 1. 1) Thanks for your suggestion. We had two independent runs and used 10 seeds to
evaluate each methods to get the max average rewards. For example, in Table 1, the standard deviation for KTM-DRL,
Ideal, TD3-MT, SAC-MT, SharedNet, G-Surgery, and A2C-MT on the task HalfCheetahSmallTorso are 10348±476,
8743±547, 7898±347, 7805±899, 4140±907, 7189±608, 2276±46.14, respectively. 2) Since KTM-DRL learns from
task-specific teachers, the performance of the teachers will affect the performance of KTM-DRL. KTM-DRL will fail
to find a good control policy if the teachers suffer from performance degradation in their tasks.

2. We will release all the training and evaluation code for the purpose of reproducing the results.

[Meta-Review · NeurIPS 2020]

The work proposed a simple multi-task RL approach to continuous control through two-stage training, an offline stage with policy distillation and an online stage to fine-tune the meta-policy with online transitions collected from interacting with actual environment. The paper overall is well written and easy to understand. Reviewers appreciate the extensiveness of the experiments and ablation studies demonstrating the effectiveness of the proposed approach. It is encouraging to see the simple framework achieve significant boost over state-of-the-art multi-task RL approach.